# A Scoping Review of the Evidence on Prevalence of Feline Upper Respiratory Tract Infections and Associated Risk Factors

**DOI:** 10.3390/vetsci11060232

**Published:** 2024-05-22

**Authors:** Uttara Kennedy, Mandy Bryce Allan Paterson, Ricardo Soares Magalhaes, Thomas Callaghan, Nicholas Clark

**Affiliations:** 1UQ School of Veterinary Science, The University of Queensland, Gatton, QSD 4343, Australia; r.magalhaes@uq.edu.au (R.S.M.); t.callaghan@uq.edu.au (T.C.); n.clark@uq.edu.au (N.C.); 2RSPCA Queensland, Animal Care Campus, Wacol, QSD 4076, Australia; mpaterson@rspcaqld.org.au

**Keywords:** scoping review, feline, respiratory, animal shelter, cat flu, herpesvirus, calicivirus, *Mycoplasma felis*, *Bordetella bronchiseptica*, *Chlamydia felis*

## Abstract

**Simple Summary:**

Feline upper respiratory infections (URI) are often endemic in animal shelters. Infections lead to increased morbidity, longer duration of stay at the shelter and, subsequently, reduced welfare and quality of life of affected cats. They impede the shelter’s ability to use already stretched resources and space optimally and can also result in increased rates of euthanasia of seriously affected cats. Despite the clear association of infection rates and shelter animal welfare, data on the prevalence of disease and the relative contribution of potential risk factors remain unclear, and no systematically synthesised research exists. In this scoping review, we provide a comprehensive description of the literature on feline URI in multiple contexts and explore whether the literature lends itself to be systematically synthesised. We describe patterns in spatial locations of studies, the range of pathogens and diagnostic tests, cohort characteristics and the findings of risk factor analyses. Assessing the impact of risk factors has the potential to alleviate the severity of disease, especially in shelters; however, the results were not easily pooled as the studies used inconsistent approaches. We present recommendations for ongoing epidemiological research on feline URI to provide a more structured framework and define research questions for future systematic reviews.

**Abstract:**

Feline upper respiratory tract infections (URI) are of concern, especially in animal shelters. This scoping review identifies epidemiological literature on URI as caused by feline herpesvirus (FHV), feline calicivirus (FCV), *Chlamydia felis*, *Mycoplasma felis* and *Bordetella bronchiseptica*. Four databases were searched, studies were screened, and data were extracted on a standardised template. We described patterns in spatial locations of the studies, the range of pathogens and diagnostic tests, cohort characteristics and the findings of risk factor analyses. A total of 90 articles were selected for final data extraction. There was diversity in sampling methods, precluding quantitative meta-analysis of prevalence reports. FHV was most frequently studied (*n* = 57/90). The most popular sampling site was conjunctival swabbing (*n* = 43). Most studies (*n* = 57) used polymerase chain reaction (PCR) to confirm diagnosis. Approximately one-third (*n* = 32/90) of the studies included sheltered felines. This review explores the current state of knowledge on the epidemiology and risk factors of feline URI. Assessing the impact of risk factors has the potential to alleviate the severity of disease, especially in shelters; however, the results were not easily pooled as the studies used inconsistent approaches. We present recommendations for ongoing epidemiological research on feline URI to provide a more structured framework and define research questions for future systematic reviews.

## 1. Introduction

Feline upper respiratory infection (URI) is a severe problem in cat populations around the world. It is especially of concern in animal shelters, where animals often face poor air circulation, high turnover of populations, concurrent comorbidities and previous malnourishment [1]. Infections lead to increased morbidity, and evidence suggests that reductions in infection burden are associated with improvements in animal welfare. For example, a study from a large shelter in Canada found that, when the welfare of animals was improved, animals demonstrated a decreased incidence of URI as well as lower stress and higher immunity indicators [2]. Due to the high incidence of URI in shelters, it is common for shelters to have dedicated isolation facilities for affected animals [3,4] or quarantine facilities for incoming animals [5], thus impeding these shelters’ ability to use resources and space optimally [6]. The personal experience of the authors (UK and MP) from long-term employment at a large Australian shelter has been similar with 75% of feline wards in the shelter dedicated to URI isolation. Infections also result in an increased rate of euthanasia of affected cats [7,8]; one study cited URI as the second highest reason for euthanasia in American shelters [9]. Similarly, Dinnage, Scarlett [3] found that, in a large shelter in America, one-third of kittens and two-thirds of adult cats were euthanised due to showing signs of URI. More recently, while rates of URI were not specifically cited, RSPCA Australia reported that just over 20% of felines arriving at shelters were euthanised with approximately 43% of these being for medical or infectious reasons [10]. While we were unable to extract data on the specifics of these medical/infectious reasons, it has been the authors’ experience that many of these were due to feline URI, especially where the clinical picture was further complicated by concurrent comorbidities (skin infections, chronic systemic diseases, FIV status).

Feline herpesvirus (FHV) and feline calicivirus (FCV) are the most common pathogens causing URI, but *Chlamydia felis* (*C. felis*), *Mycoplasma felis* (*M. felis*) and *Bordetella bronchiseptica* (*B. bronchiseptica*) are also reported to be involved. Common clinical signs include serous, mucoid or mucopurulent nasal discharge; sneezing; dyspnoea; conjunctivitis and ocular discharge; and ulcerations of the lips, tongue, gums, or nasal planum [1]. Infected cats often become chronic carriers and can continue to shed the viruses for their lifetime [11]. This chronicity of the disease can cause permanent scarring of nasal passages, leading to chronic sinusitis and recurrent bacterial infections of the upper respiratory tract [7].

A preliminary literature search by the authors found that several population-level studies evaluated the associations of feline URI with various risk factors, reported on disease frequency and tested the feasibility of different diagnostic and sampling methods. Evidence suggests that age, gender, breed and presence of clinical signs are likely predictive of URI in some contexts [12,13,14], but the generality of these associations is not known. High-stress housing (e.g., group housing, litter trays in proximity to living areas, high foot traffic and multiple species within hearing and smelling distance) is noted for its potential contribution [15]. Other risk factors include frequent relocation of cats between and within shelters [16], space per cat [6] and quality of human interactions [2,4]. Despite the clear association between URI and shelter animal welfare, data on the prevalence of the disease and the relative contribution of potential risk factors remain unclear, and no systematically synthesised research exists.

Systematic reviews have long been considered the most “reliable and comprehensive assessment” of empirical evidence [17]. They have been a part of human health literature for over 40 years [18] and have come to represent the pinnacle of evidence-based medicine. Systematically synthesised information on feline URI pathogen prevalence, its correlation with clinical signs, effective methods of diagnosis and prevention of spread through risk factor analyses would feed directly into management strategies to improve the welfare of affected animals, especially in high-density settings like animal shelters. It, therefore, becomes an important question to ask why a systematic assessment of feline URI occurrence has not been carried out and whether it is feasible to do so. Through our preliminary literature search, it was anecdotally evident that there was heterogeneity in study approaches: case definitions, diagnostic methods and sampling techniques varied between studies. In light of these findings, we deemed it appropriate to conduct a scoping review of the extent and diversity of evidence on this disease. Arksey and O’Malley [19] describe at least four reasons a scoping review may be conducted: (1) to examine the range (diversity) and characteristics of studies in a particular area, (2) to assess the feasibility of systematically synthesising results from these studies, (3) to summarise and disseminate research findings and (4) to identify gaps in the existing literature. They also present a methodological framework for conducting a scoping review; we use the steps of this framework with a view to addressing the four areas. The methodological steps include identifying the research question, identifying relevant studies, study selection, charting the data and collating and summarising the results. While our review does not include bias assessment or complete synthesis of research findings, it provides a comprehensive description of the literature on frequency data reported for feline URI in multiple contexts and explores whether the literature lends itself to be systematically synthesised. Our objective was to conduct a scoping review of observational studies on feline URI. The research questions were the following: “What is the current body of evidence on the presence of feline URI and its association with potential risk factors? Can this information be systemically synthesised?”.

## 2. Materials and Methods

This review uses the five-step process as described by Arksey and O’Malley [19].

**Define the research question:** Our aim was to gather and describe all relevant peer-reviewed observational research on feline URI that reported on measures of disease frequency. All measures of disease frequency (point prevalence, period prevalence, cumulative incidence and incidence rate) for any of the five pathogens of feline URI were included as long as they were reported in the domestic cat (*Felis catus*).

**Identify relevant studies:** We searched four electronic databases: Medline (1966–present), PubMed (1809–present), Science Direct (1823–present) and Web of Science (1900–present) [20,21,22,23]. The searches were conducted using a combination of search strings for each database (Table 1). Our goals were to capture all studies that mentioned any of the five pathogens of feline URI in an epidemiological capacity. We included any term that may be used to describe measures of disease frequency in a population (e.g., incidence, prevalence, detection). The search strings were constructed with Boolean operators and tuned by using combinations of keywords until we obtained consistent search returns of relevant articles. Adding keywords beyond this optimal combination did not yield any more relevant articles. To ensure maximal inclusion, no specific date range or language filters were applied. The searches were completed between July 2021 and August 2021. The results of each database search were stored in Endnote (Version X9.3.3), and duplicate articles were removed according to the methods described by Falconer [24].

**Study selection:** Study selection was conducted in two phases: first, by screening the titles and abstracts of all unique articles obtained through the database searches; and second, by reading the full text of each article selected in the preceding step. For both steps, screening for inclusion was performed using pre-defined selection criteria (Figure 1). Titles and abstracts were manually screened by the main author (U.K.), and if feline URI or one or more of the five pathogens implicated in feline URI were mentioned in a population-based observational context, the article was included. Case reports, experimental animal models or studies not reporting on domestic feline species (*Felis catus*) were excluded. To validate the selection process, every 8th article was chosen to obtain two sets of 31 articles each. These were sent to two independent verifiers together with the pre-defined selection criteria. The results were compared for agreement or discrepancy; if articles were found to be erroneously included or excluded, these were re-categorised by consensus between the verifiers and authors.

After completing title and abstract screening, the primary author (U.K.) manually checked the full text of each selected article for eligibility using the same pre-defined criteria in Figure 1, resulting in a final list of articles included for the review. Studies that had primary aims other than reporting on disease frequency but did have some frequency measures in their results were included for data extraction. Full-text manuscripts were obtained through University of Queensland’s institutional access. Where full-text articles were not found through institutional access, they were obtained by request via the university’s inter-library loan system. In one case, a full-text manuscript was not obtainable, and another two full-text articles were in a foreign language with English abstracts: for these three studies, the abstracts were used for data extraction. An automatic email alert system was set up with each database, and any additional articles meeting the relevance criteria were included. The bibliographies of all included articles were manually scanned to check for additional articles to add to the final list. No additional articles were included after February 2022.

**Charting the data:** A standardised template for data capture was created in Microsoft Excel (2004). The primary goal was to capture information on the spatial distribution of studies and to determine whether the reported frequency data for each pathogen could be spatially or temporally compared or pooled. The secondary goal was to collect data on the most commonly studied pathogens, diagnostic methods and sampling techniques that were used for identifying diseased animals and the characteristics of the sampled cohorts. While the initial fields for the template were determined prior to data extraction, the template expanded as fields were added as the review progressed. The final template included the following fields: location, date and time range of each study, primary aim, secondary aim, type of population, diagnostic and sampling methods, risk factors, health status of sampled animals, number and types of pathogens studied and measure of disease frequency.

All articles were categorised or described as per pre-defined questions for each field on the final data extraction template (Appendix A). Where primary and secondary aims of the studies were not explicitly stated in the introduction, the studies were categorised by determining the main focus of the reported results within that study. The risk factors studied were too numerous and diverse to categorise, and these were listed as a general field. The categories for each field are shown in Table 2. For studies that did not specify the year or date when the study was conducted, the year of publication was recorded as a substitute for the date of study. For consistency within our review, where studies have conducted laboratory testing and used terms such as ‘test positivity’, ‘pathogen shedding rate’, ‘detection rate’ and ‘recovery rate’, we interpret them to mean ‘prevalence’. Amongst the studies that did not conduct laboratory tests, we define the term ‘prevalence’ for the number or percent of animals with clinical signs. ‘Clinical signs’ denotes the presence of visible ocular, nasal or other respiratory signs that indicate the likelihood of URI infection.

After data extraction was completed, two groups of 5 articles each were made. The articles, together with blank fields within the data extraction template, were sent to two separate verifiers with the pre-defined questions for each field (Appendix A) and the format of recording (binary vs. descriptive). The validators populated the blank fields in the spreadsheet by reading the full text of each article, and the results were matched for validation. Discrepancies were evaluated, and modifications to the data spreadsheet were made by mutual consensus.

**Collating and summarising results:** Once all data were collected, we used summary statistics and frequency tables and plots to describe patterns in the spatial locations of the studies, types of pathogens surveyed, types of diagnostic tests used, cohort characteristics and findings of risk factor analyses.

## 3. Results

A total of 243 unique articles were identified from the four databases. After title and abstract screening, 95 articles were included for full-text evaluation. Three further articles were added from hand searches and email alerts. Eight articles were eliminated after full-text screening, leaving 90 articles for final data extraction and analysis (Figure 2). The earliest study was carried out in 1971 with an increasing number of studies over time, with most studies (*n* = 65/90) between 2001 and 2020. The largest number of studies were performed on animal populations in Europe (*n* = 39/90) (Figure 3). A majority of studies either explicitly stated or were determined by the authors (from the focus of reported results) to have prevalence reporting as one of their primary goals (*n* = 72/90) (Table 3). However, there was diversity in sampling and statistical methods as well as terminology used by these studies, precluding any quantitative meta-analysis or aggregation of prevalence reports from the studies included in this review (Table 4). FHV was the most frequently studied pathogen (*n* = 57/90), followed by FCV (*n* = 47/90) as shown in Table 5. The most common pair of pathogens studied together were the two viruses (Appendix A). Since some studies looked at multiple pathogens, these numbers are different from the total number of studies. Just under half of the studies looked at a single pathogen (*n* = 43/90), and three studies did not conduct laboratory testing for any of the pathogens but relied on case definitions based on clinical signs (Table 6). A total of 23 out of 90 studies had some mention of the presence or extent of coinfections.

The most popular sampling site for URI infections was conjunctival swabbing (*n* = 43), followed by oropharyngeal swabbing (*n* = 38) and nasal swabbing (*n* = 21). The most common combination of sample sites was oropharyngeal and conjunctival swabbing (*n* = 22). Other samples included serum, conjunctival scrapings, bronchoalveolar lavage fluid, corneal biopsies and tissue biopsies. Most studies (*n* = 57) used various forms of polymerase chain reaction (PCR) to confirm diagnosis. A breakdown of test type by pathogen is shown in Figure 4.

We found that 45/90 studies investigated the association between test positivity and presence of clinical signs. While several studies reported only the presence/absence of significance to assess the relationship between clinical signs and test positivity, eight studies provided effect estimates that could potentially facilitate a meaningful synthesis. However, the studies varied in design and reported associations in a heterogenous manner, as demonstrated in Table 7.

### 3.1. Studies That Included Animal Shelters

Despite the great burden that feline URI puts on animal shelters and the quality of life for affected cats, only a little over one-third (*n* = 32/90) of the studies included sheltered felines, of which only six reported on the prevalence of all five pathogens. These six studies all used PCR as one of their diagnostic techniques, and all of them used at least two sampling sites, the most common combination being conjunctival and oropharyngeal swabs. The range of reported prevalence amongst these studies was wide (e.g., FHV 2%–94%). It was encouraging to note that 23 of 32 studies reported on the association with potential risk factors. Overall, the most studied risk factors were demographic (age, gender, vaccination status) and environmental factors (housing, source). We investigated these studies to determine if common risk factor associations emerged and whether these results could be used for potential future meta-analyses.

#### 3.1.1. Age as a Risk Factor in Shelters

Of the studies including shelters, ten analysed the impact of age on URI, but there was a wide diversity of detection and statistical methods used to assess these impacts. Three studies found that animals younger than 12 months showed a higher rate of clinical signs [9,16,25]. One study found that adults between 12 and 36 months showed a greater number of positive PCR test results for FHV [12], while Dinnage, Scarlett [3] found that geriatric cats over the age of 11 years showed a significantly higher prevalence of clinical signs. Of the remaining studies, two found no significant difference in shedding between age groups [31,32], two assessed the impact of age on serum antibody titres [33,34], and one study only reported on the effect of age in relation to vaccination status [26]. Table 8 summarises the association of age with pathogen burden or clinical signs in shelters.

#### 3.1.2. Environmental Risk Factors in Shelters

The studies that explored the correlations between shelter practices and feline URI were also multifactorial and complex. They indicate that cats with more space, less disturbance, less movement and more environmental enrichment were less likely to show clinical signs; however, specific risk factor descriptors were diverse. As an example, a study conducted in the USA found a significant positive linear relationship between time in transport and the presence of clinical signs [16]. Wagner, Kass [6] found the presence of double-compartment housing and a lower frequency of movement of cats between cages in the first week was associated with lower rates of clinical signs. Table 9 presents a descriptive summary of the studies in shelter settings and the various environmental factors that were found to be associated with clinical signs or pathogen shedding.

#### 3.1.3. LOS as a Risk Factor in Shelters

Four studies analysed the association of LOS with the prevalence of URI in shelters. However, two of these did not conduct laboratory tests and used only the presence of clinical signs to report on prevalence estimates, precluding a meta-analysis of the four studies. While all studies found a positive association between the development of clinical signs and LOS, not all found the association to be linear; some studies demonstrated a plateauing or reduction in prevalence after a few weeks. Dinnage, Scarlett [3] calculated the cumulative probability of developing clinical signs over time and found that it increased from 5% in the first two days to over 80% after two weeks. Similarly, Courkow, Lawson [32] found the cumulative risk of developing both clinical signs as well as pathogen shedding to steadily increase for up to 30 days of stay in the shelter. Bannasch and Foley [9] compared a diverse range of animal care facilities and found that all the facilities showed a positive association between LOS and clinical signs, but the levels were higher in higher-density facilities, staying elevated in traditional shelters but reducing after two weeks in no-kill facilities.

## 4. Discussion

This scoping review comprehensively describes the literature on feline URI in multiple contexts. Specifically, we give an overview of the distribution of studies conducted across the world as well as the diversity in disease frequency reporting, sampling methods and types of cohorts included for study. Additionally, we discovered the most commonly studied pathogens, diagnostic methods and risk factors. We investigated the consistencies and inconsistencies found between papers studying the same risk factors within similarly housed cohorts (animal shelters) around the world. Identifying trends and commonalities as well as heterogeneity among studies can guide future research so that more comprehensive systematic comparisons can eventually be made.

**Prevalence estimates:** A simple but useful goal for research synthesis is to estimate the prevalence of disease across different contexts [37]. By scrutinising how prevalence was reported, we found that there was statistical and terminological variation that would impede any such synthesis with studies using a range of indicators, including percent period prevalence, number or percent of positive tests, prevalence of antibody titres and incidence. Dinnage, Scarlett [3] and Courkow, Lawson [32] conducted longitudinal studies that reported cumulative probabilities, while Wong, Kelman [38] conducted a database search for diagnoses, and prevalence was reported using the number of cases identified. The result was that denominators varied between studies, depending on study type and whether only symptomatic animals were included, as in Becker, Monteiro [39] or only healthy animals as in Aziz, Janeczko [16], likely influencing pathogen prevalence results and further complicating any attempt at research synthesis. At least 29 studies mentioned the inclusion of a specifically chosen healthy control group; however, others did not mention whether the ‘healthy’ cohort was included by random selection or as matched controls. Often, separate prevalence estimates were reported at various points in time [5], for various types of facilities [40], for different geographical regions [41] or through multiple diagnostic methods or sample sites [36,42,43]; in the latter case, prevalence estimates are likely to be dependent on the reliability and validity of diagnostic methods used to obtain them [37].

Some studies used protective antibody titres to determine the extent of circulating pathogens; however, the association of serum antibodies with clinical signs was inconsistent. While some studies found a positive association between the presence of serum antibodies and clinical signs [34], others found no significant association [33,44]. Since antibody presence is presumed to be linked with developed immunity through either natural exposure or vaccination [45] rather than ongoing infection, we query the value of serum antibody testing as either diagnostic or indicative of true prevalence of disease.

**Diagnostic Methods:** Understanding the variation in the performance of different diagnostic tests is another key focus of research syntheses [46]. Amongst the studies that compared diagnostic tests, many did not report the association of results with clinical signs. PCR results, especially, need to be interpreted with caution, as positive tests do not necessarily indicate active infection; routine vaccinations as well as viral shedding by latently infected yet clinically healthy animals can lead to positive PCR results [40,47,48,49], which may lead to unnecessary isolation, treatment or even euthanasia. In real-world settings, especially shelters, a presumptive diagnosis based on clinical signs can often be adequate, as there is considerable overlap of clinical signs as well as treatment options between pathogens [45]. To investigate the value of laboratory testing, further research into the association between test positivity and clinical signs is needed. While some studies did provide effect estimates that could potentially facilitate a meaningful synthesis, the heterogeneity in study design and reporting would make this challenging.

**Co-infections:** Co-infections are common and can play a role in lowering immunity and increasing susceptibility. Several studies reported on co-infection as a single percent of the total sample for example, Becker, Monteiro [39]) without a meaningful breakdown of the specific pathogens that were co-detected. Many studies also included several common non-respiratory feline pathogens: Abayli, Can-Sahna [12] found that FHV and FCV often co-occurred in animals with feline immunodeficiency virus (FIV) and feline panleukopenia virus (FPLV), while Bayraktar and Yilmaz [50] reported on the frequency of FHV occurring with FIV and feline leukemia virus (FeLV). It is likely that the presence of one pathogen influences the behaviour of the others, especially in high-density settings [25], yet these relationships were not assessed by any of the studies. While some pair-wise co-infection estimates were reported (for example, Rampazzo, Appino [51] reported that co-infection of FHV and *C. felis* occurred in 7% of sampled cats), reports on the same combination of pathogens were few.

**Age as a risk factor:** It is likely that very young kittens are especially susceptible before they have had a chance to be vaccinated due to waning maternal antibodies [16]. Dinnage, Scarlett [3] argue that geriatric animals are more likely to show clinical signs due to having had a greater chance of being latently infected during their lifetime and being susceptible to reactivation due to old age and stress. We found that studies demonstrated a weak trend towards younger and older animals having more association with pathogen burden or clinical signs.

**Environmental risk factors in shelters:** In general, the studies included in this review indicate that cats with more space, less disturbance, less movement between enclosures or shelters and more environmental enrichment were less likely to show clinical signs; however, specific risk factor descriptors were diverse. A study conducted in the USA found a significant positive linear relationship between time in transport and the presence of clinical signs [16]. Bannasch and Foley [9] found that the prevalence of clinical signs between shelters varied considerably depending on the shelter’s facilities, such as housing design, the presence of isolation cages and the average length of stay for animals at each shelter. Wagner, Kass [6] found the presence of double-compartment housing and a lower frequency of movement of cats between cages in the first week was associated with lower rates of clinical signs. Feline housing in close proximity to dogs was associated with increased rates of infection [9,11], and higher-density settings such as shelters showed a higher rate of infection when compared with private households [40] or free-roaming populations [52]. It is likely that smaller, single-storied cages can increase stress levels by preventing cats from being able to express natural behaviours, such as full stretches, running, jumping, rolling and being able to move more than a few steps away from litter areas. Smaller cages are also less likely to house multiple animals and have less environmental enrichment and opportunity for social interaction [15].

**Length of Stay (LOS) as a risk factor in shelters:** An animal’s LOS in a shelter is emerging as an important metric in assessing a shelter’s ability to successfully rehome animals [53]. It is likely that, in all studies, any reduction in prevalence after two weeks could be due to cats being euthanised within the first few days of showing clinical signs [9] or the low numbers of cats that tend to remain in shelters beyond 3 weeks [3]. It has been reported that prospective adopters prefer younger, lighter coloured males [36], resulting in the less preferred animals staying at shelters for longer periods of time. The longer an animal stays in a shelter, the greater its stress and exposure to pathogens and subsequent chances of becoming infected with common endemic diseases [7]. The time needed for subsequent medical treatment then increases overall LOS, or the animal is euthanised if the shelter does not have the necessary resources needed for an extended stay and treatments [53,54,55]. Both outcomes are unfavourable for shelters in terms of resource management, animal welfare outcomes, public perception and staff morale.

Quarantine facilities are often based on the assumption that incoming animals are likely to be carrying infectious diseases and that their spread can be limited through mandatory quarantine. However, Aziz, Janeczko [16] discovered that a high proportion of animals were latent carriers for FHV, which is activated by stressful events (such as confinement for quarantine). In such cases, mandatory quarantine periods increase total LOS, prolonging shedding and resulting in increased overall morbidity [3,32]. Most studies suggested that stress management, routine health surveillance, isolation of symptomatic animals and thorough cleaning and disinfection of cages and common areas can be more effective mitigation tools instead of mandatory quarantine.


**Gaps in knowledge and recommendations for future epidemiological research:**


There is a dearth of systematic evidence summaries on feline URI in the literature. However, such summaries, even if present, will have inherent limitations if study designs, variation amongst studies, within-study biases and reporting biases are not considered and assessed [37,56]. Some of the challenges we discovered in this review include (1) inconsistent definition of exposures; (2) inconsistent definition of outcome measures (e.g., prevalence as determined by clinical diagnosis vs. positive laboratory test); (3) lack of distinction between carrier states, active infection and protective immunity (e.g., asymptomatic positive laboratory test vs. diagnosis based on clinical signs vs. protective antibody titres); (4) lack of clarity on denominators (e.g., method of selection of controls not outlined); (5) no uniform definition of clinical diagnosis as outcome/risk factor (e.g., history of respiratory disease vs. chronic gingivostomatitis vs. sneezing vs. ocular signs), precluding pooling of results for a single definition of URI; (6) diversity in sampling technique and diagnostic methods (e.g., oropharyngeal swabs vs. corneal biopsy; ELISA vs. PCR testing), which can reflect in validity and accuracy of results; and (7) diversity in definition of environmental condition as a risk factor (e.g., group housing or single housing vs. double-storey or single-storey housing, precluding pooling of results for a single definition of housing as a risk factor).

While it is beyond the scope of a review like this one to extract outcome data or make quality assessments of included studies [57], we were able to identify trends and commonalities between studies as well as inconsistencies and heterogeneity. Taking into consideration the extent of heterogeneity between study designs, definitions of exposures and outcomes, study aims and statistical methods and reporting, future systematic appraisals with broad research questions (e.g., mapping the prevalence of feline URI disease in cat populations around the world) are not realistic. Instead, a research query with a narrow focus can lead to a more meaningful evidence synthesis. For example, the results on the prevalence of clinical signs need to be differentiated from those on the prevalence of pathogen shedding. For studies synthesising the prevalence of clinical feline URT disease, a clear definition of the disease syndrome should include all of the commonly observed signs, such as nasal discharge, ocular discharge, conjunctivitis, corneal ulceration, coughing, sneezing and oral ulceration.

Since the pathogenesis of URI varies depending on the environment, for future systematic reviews, we recommend differentiation between study populations (e.g., cats from a shelter vs. owned cats) for data synthesis. Likewise, diagnostic methods and sampling techniques differ from each other in accuracy and clinical significance, and frequency data should be weighted accordingly before being pooled. For synthesising results from risk factor analyses, we recommend a clinically relevant approach: a focus on environmental factors that are known to induce stress and are easily modifiable rather than those that cannot be changed or controlled (e.g., housing density vs. seasonal variation). These include (but are not restricted to) LOS, time spent in transport, physical interaction with other cats, size of enclosures (floor space), multi-level cages, proximity to dogs and positive human interaction.

## 5. Conclusions

Through this scoping review, we explore the current state of knowledge on the epidemiology, diagnosis and risk factors of feline URI. URI has been studied in different types of populations (stray, privately owned, colonies, shelters, etc.) around the world, but it specifically remains a severe welfare problem in shelters. While several individual studies found that PCR tests yield higher positivity rates when compared to other diagnostic methods such as viral or bacterial isolation, the relevance of a positive PCR test to clinical signs warrants further research. There was also agreement that oropharyngeal and conjunctival swabbing yield the highest recovery rates when compared to other sites. Approximately one-third of the studies included shelter facilities, and many of these studied the association of disease with demographic and environmental risk factors. Assessing and modifying the impact of stress-inducing risk factors have the potential to alleviate much of the severity of disease in shelters. However, the results from risk factor analyses can only be pooled for meta-analysis if studies use consistent definitions and approaches. We present some guidelines and recommendations for ongoing epidemiological research on feline URI, which will not only provide a more structured framework for individual studies but will also help define specific research questions for future systematic reviews.

## Figures and Tables

**Figure 1 vetsci-11-00232-f001:**
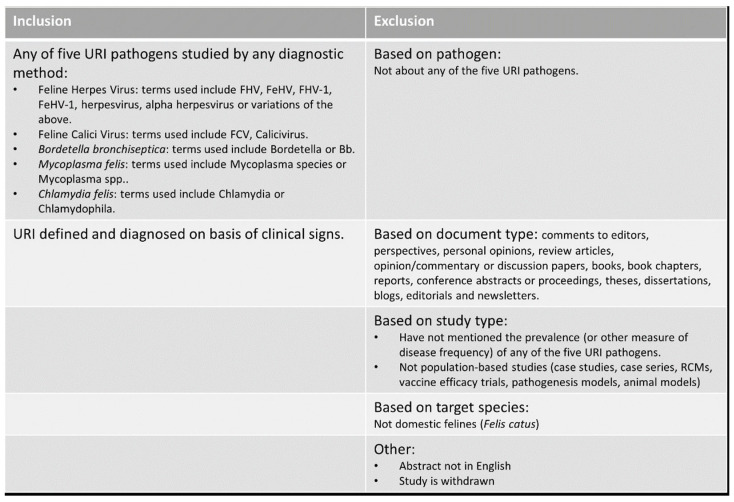
Inclusion/exclusion criteria for manual title–abstract and full-text search.

**Figure 2 vetsci-11-00232-f002:**
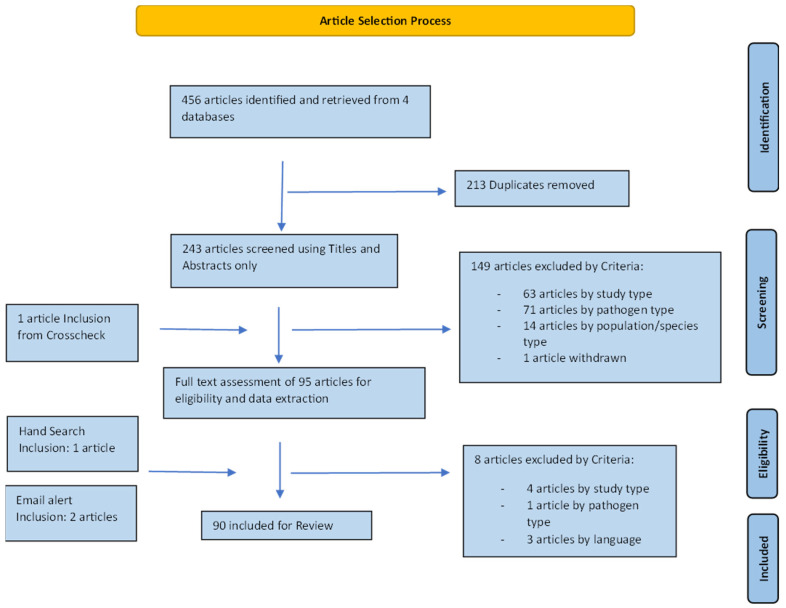
Schematic of the article selection process.

**Figure 3 vetsci-11-00232-f003:**
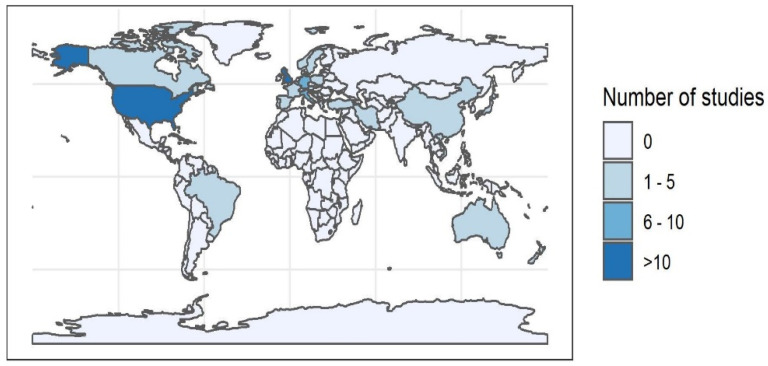
Number of studies conducted globally.

**Figure 4 vetsci-11-00232-f004:**
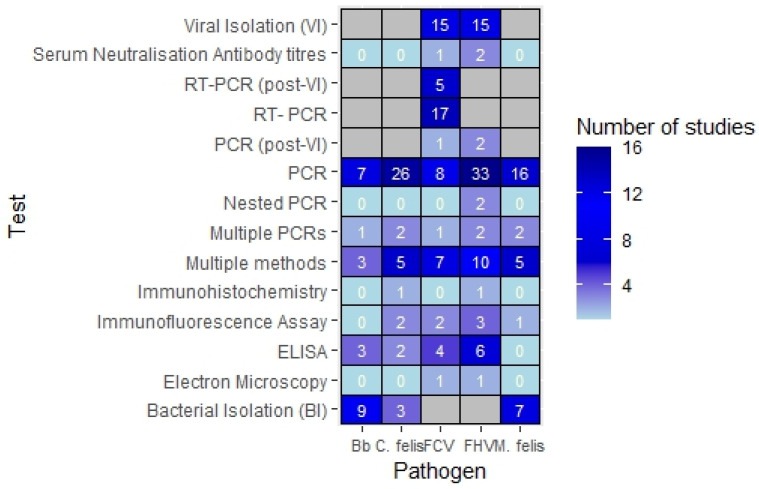
Number of studies by type of tests conducted per pathogen.

**Table 1 vetsci-11-00232-t001:** Database search strings and number of unique results.

Database	Search Strategy	Number of Results
**PubMed**	**Title:** ((feline[Title] OR cat[Title] OR cats[Title]) AND (prevalence[Title] OR incidence[Title] OR detection[Title] OR surveillance[Title] OR occurrence[Title] OR epidemiol*[Title])) AND (URI[Title] OR respiratory[Title] OR URTD[Title] OR FHV[Title] OR FeHV[Title] OR “FHV-1”[Title] OR “FeHV-1”[Title] OR herpes*[Title] OR calici*[Title] OR chlamyd*[Title] OR bordetella[Title] OR mycoplasma[Title]) AND (journalarticle[Filter]) **Refine by Document Type:** Articles	134
**Web of Science**	**Title:** TI = (feline OR cat OR cats) AND TI = (prevalence or incidence or epidemiolog* or surveillance or occurrence or detection) **Abstract:** AB = (FHV or FeHV or “FHV-1” or “FeHV-1” or herpes* or calici* or bordetella or chlamyd*) OR AB = (flu or URI or FURTI or URTD) **Refine by Document Type:** Articles	122
**Medline**	**Title:** TI (feline or cat or cats) AND TI (epidemiology OR prevalence OR detection OR incidence OR identification OR surveillance OR occurrence) **Abstract:** AB (flu or herpes or herpesvirus or URI or URTD or FURTI or URTI or calici or calicivirus or FHV or FCV or FHV1 or FHV-1 or FeHV or FeHV-1 or mycoplasma or chlamydia or chlamydophila or bordetella) **Entire Text:** TX (respiratory OR flu OR shelter OR rescue OR “URI” OR “URTD” OR chlamydophila OR herpesvirus OR calicivirus) **Limiters**—Scholarly (Peer Reviewed) Journals **Expanders**—Apply related words; Apply equivalent subjects **Search modes**—Boolean/Phrase	152
**Science Direct**	**Title:** (Feline OR cat) AND (epidemiology OR prevalence OR detection OR incidence OR identification OR surveillance OR occurrence) **Title, abstract, keywords:** “FHV” OR “FeHV” OR “FHV1” OR “FeHV1” OR Calici OR Herpes OR chlamydia OR mycoplasma OR respiratory **Entire text:** respiratory OR flu OR shelter OR rescue OR “URI” OR “URTD” OR chlamydophila OR herpesvirus OR calicivirus **Research Articles ONLY**	48

**Table 2 vetsci-11-00232-t002:** Data extraction fields for categorisation.

Field	Category
Location (continent and country)	Descriptive
Length of study	Numeric (months)
Primary aim	Descriptive: prevalence, diagnostic evaluation, sample site evaluation, sequencing, etc.
Secondary aim	Descriptive: prevalence, diagnostic evaluation, sample site evaluation, sequencing, etc.
Pathogen	Binary indicator: FHV, FCV, *Mycoplasma felis*, *Chlamydia felis*, *Bordetella bronchiseptica*
Diagnostic method	Binary indicator: PCR, viral isolation, bacterial isolation, ELISA, other (descriptive)
Sample type	Binary indicator: oropharyngeal cytobrush, oropharyngeal swab, conjunctival swab, nasal swab, clinical signs, other (descriptive)
Sample size	Numeric
Population	Binary indicator: shelter, owned, other (descriptive), combination
Health status of sampled animals	Binary indicator: healthy, symptomatic, both
Risk factors	Descriptive (list)
Disease frequency	Numeric or descriptive based on reporting

**Table 3 vetsci-11-00232-t003:** Number of studies by primary goal/aim of study.

Terminology for Measure of Frequency Estimate	Number of Studies
% animals/samples positive	34
Prevalence	32
Number of animals/samples positive	9
Multiple prevalence estimates over time	2
Crude IDR, Prevalence and Cumulative Probability	2
Prevalence Range	1
Incidence	1
Period Prevalence	1
Combination of above terms within study	8

**Table 4 vetsci-11-00232-t004:** Diversity of reporting terminology for measures of disease frequency.

Primary Goal/Aim	Number of Records
Prevalence reporting	72
Diagnostic evaluation	7
Sequencing	1
Sample site evaluation	2
Cumulative incidence	1
Association with bronchitis and asthma	1
Association with ocular symptoms	2
Association with oral symptoms	2
Association with feline lower urinary tract disease	1

**Table 5 vetsci-11-00232-t005:** Number of studies reporting on multiple pathogens vs. single pathogen.

Number of Pathogens Studied	Number of Studies
5	7
4	4
3	10
2	23
1	43
0 (based on clinical signs only)	3

**Table 6 vetsci-11-00232-t006:** Number of studies reporting on each pathogen.

Pathogen	Number of Studies
FHV-1	57
FCV	47
Bb	15
*M. felis*	20
*C. felis*	31

**Table 7 vetsci-11-00232-t007:** Studies reporting statistical association between clinical signs and pathogen shedding (CGS = chronic gingivostomatitis, LGSC = lymphoplasmacytic gingivitis stomatitis complex).

Article	Pathogens Tested	Study Design	Association Reported
[11]	FHV, FCV	Cross-sectional	Assoc. with multiple individual respiratory signs
[13]	FCV	Cross-sectional	Assoc. with CGS
Binns et al. (1999) [25]	*B. bronchiseptica*	Cross-sectional	Assoc. with current and past respiratory disease
[26]	FHV, FCV	Case Control	Sample selection method not specified. Assoc. with LGSC (FCV only)
[27]	FCV	Case Control	Assoc. with enteric symptoms
[28]	FHV	Case Control	Individual assoc. with sneezing and disease duration
[29]	FHV, FCV, *B. bronchiseptica*, *M. felis, C. felis*	Prospective Cohort	Individual assocs. with past and future salivation and nasal discharge
[30]	FCV	Cross-sectional	Individual assocs. with several respiratory signs

**Table 8 vetsci-11-00232-t008:** Studies reporting association of age as a risk factor (w = weeks, m = months, y = years).

Article	Pathogens (Diagnostic Methods)	Age Brackets (Bold Bracket Has Significant Increased Association)	Statistical Results
Abayli et al. (2021) [12]	FHV (PCR)	<12 m, **12–24 m**, 25–36 m, >36 m	*p < 0.01*
[16]	Clinical signs only	**<5 m**, >5 m	*OR = 0.3, p < 0.01*
[9]	FHV, FCV, *B. bronchiseptica*, *M. felis*, *C. felis* (PCR, VI, BI)	**0–3 m**, 4–6 m, **7–11 m**, 12+ m, 72–96 m, 96+ m	*OR = 1.85 (1.25–2.74), p < 0.01* *OR = 2.58 (1.24–5.41), p = 0.01*
[11]	FHV, FCV (VI, clinical signs)	1–3 m, **4–11 m**, 1–3 Y, 4–7 Y, >7 Y	*OR = 2.9 (1.1–7.3), p = 0.03*
[33]	FHV, FCV (antibody titres)	<6 m, **6–11 m**, **1–5 Y**, **>5 Y**	*OR = 15.3 (1.4–392.9), p = 0.01* *OR = 74.8 (10.6–1507.5), p < 0.01* *OR = 194 (18.9–4840.6), p < 0.01*
[3]	Clinical signs only	0–8 w, 9–15 w, 4–6 m, 7–11 m, 1–4 y, 5–10 y, **11+ y**	*p < 0.05*
[34]	*B. bronchiseptica* (PCR, ELISA, VI)	Age brackets not available, older cats had increased association	*OR = 1.10 (1.01–1.19), p < 0.05*
[32]	FHV, FCV, *B. bronchiseptica, M. felis, C. felis* (PCR)	**6–12 m**, 1–8 Y, **>8 Y**	*OR = 1.49 (0.67–3.31), p = 0.33* *OR = 1.66 (0.78–3.55), p = 0.19* *Association with clinical signs, not pathogen shedding.*
[26]	FHV, FCV (PCR, VI)	Continuous variable, used as 2-way interaction with vaccination status	na
[31]	*B. bronchiseptica* (BI)	<6 m, >6 m	*p > 0.1*

**Table 9 vetsci-11-00232-t009:** Studies assessing environmental risk factors in shelter environments (IRR = incident rate ratio, LOS = length of stay).

Reference	Country	Environmental Risk Factor Assessed (Bold Indicates Significance)	Associated Outcome
Aziz et al. (2018) [16]	USA	**LOS, time in transport**	Clinical signs
[9]	USA	**LOS, shelter type, isolation in shelter**	Clinical signs and pathogen shedding
[33]	USA	**Environment history prior to relinquishment**	Serum antibody presence
[3]	USA	**LOS, environment history prior to relinquishment,** isolation in shelter	Clinical signs
[5]	USA	**Concurrent canine infection**	Pathogen shedding
[32]	Canada	**LOS**	Clinical signs and pathogen shedding
[35]	Japan	**Treatments administered**	IRR of clinical signs
[29]	Japan	**LOS, treatments administered, housing type**	Recurrent clinical signs and/or pathogen shedding
[36]	Canada	**Season**	Pathogen shedding

## Data Availability

Not applicable.

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
