# Peer review of "A Scoping Review of the Evidence on Prevalence of Feline Upper Respiratory Tract Infections and Associated Risk Factors"

_vetsci, 2024, doi:10.3390/vetsci11060232_

Round 1
Reviewer 1 Report
Comments and Suggestions for Authors
The article clearly defines its objective and formulates reasonable conclusions.
The format is a systematic review, complete both in time and databases searched.
Material and methods are very well described, including boolean search strings, as well as inclusion and exclusion criteria.
Results and discussion are clear.
The paper may be published in its present form.
In Discussion section line 391 [...less movement...] I would like to suggest to reformulate the sentence. "Less movement" might indicate: shorter time in transport, movement between cages, smaller cages (seems unlikely).
Author Response
Thank you for your comments. We have changed like 391 to make it more specific:
"less movement between enclosures or shelters"
Reviewer 2 Report
Comments and Suggestions for Authors
In the study entitled “A scoping review of the evidence on prevalence of feline upper respiratory tract infections and associated risk factors” the Authors aim was to report the current knowledge on feline upper respiratory infections in multiple contexts and explore whether the literature lends itself to be systematically synthesized. Authors described the patterns in spatial locations of studies, range of pathogens and diagnostic tests, cohort characteristics and findings of risk factor analyses.
This study adequately reported the knowledge currently available in the scientific literature in the field. The subject of the work is of interest and that the topic of the manuscript falls within the journal topic. Authors rationale is worthy of investigation. However, the manuscript needs to be improved before it can be considered suitable for publication. English language should be improved throughout the text.
I suggest to avoid the use of personal form (i.e. our, we…) throughout the text.
I suggest to change/delete the words in the keywords which appear in the title also.
Throughout the text several sentences are redundant. Please check and delete the repetition.
The title is nice and the abstract adequately summarize results and significance of the study. The introduction section is well written and it falls within the topic of the study, and Authors cited appropriately bibliographic information. The section of Materials and Methods is clear for the reader and it meticulously describes the methods applied in the study. Specifically, the inclusion/exclusion criteria are well stated. However, did authors select papers in a specific time window? It is not clear. The conclusion is well written, indeed Authors well summarize the results and the significance of the study. I suggest to enhance the sentence “We present some guidelines and recommendations for ongoing epidemiological research on feline URI, which will not only provide a more structured framework for individual studies but will also help define specific research questions for future systematic reviews.” By briefly indicate the suggested guidelines.
I suggest to improve the quality of tables.
Comments on the Quality of English LanguageEnglish language should be improved throughout the text.
Author Response
We thank the reviewer for the comments. Each comment is addressed below in dot-point.
- English language should be improved throughout the text: We respectfully disagree that the language needs editing. We would like to point out that all authors are native English speakers and have published extensively in peer-reviewed international journals. We also note that 4 other reviewers stated that the language is suitable and does not need to be altered or revised.
- I suggest to avoid the use of personal form (i.e. our, we…) throughout the text: We would like to point out that the current preferred format for academic literature is to use first-person, as we have consistently done through this manuscript as well as previously published peer-reviewed articles.
- I suggest to change/delete the words in the keywords which appear in the title also: We have included key words to reflect all topics related to this article. While some of the words overlap, we have not come across any journal specifications to avoid this overlap, and we feel that the key words are selected for effective searches by readers in the future.
- Throughout the text several sentences are redundant. Please check and delete the repetition: Unfortunately, without examples, we are unsure of areas of repetition and are unable to address this comment. Since other reviewers have not commented on redundancies, we feel that leaving the manuscript in its current format is adequate.
- I suggest to enhance the sentence “We present some guidelines and recommendations for ongoing epidemiological research on feline URI, which will not only provide a more structured framework for individual studies but will also help define specific research questions for future systematic reviews.” By briefly indicate the suggested guidelines: We have elected to leave this sentence rather than provide examples of guidelines, since this would be an incomplete repetition of areas covered in detail in the previous section. The Conclusion may become too long to be accepted if we outline all guidelines again in this section.
- I suggest to improve the quality of tables: All tables are created in MS Word and are in line with journal specifications (All table columns should have an explanatory heading. To facilitate the copy-editing of larger tables, smaller fonts may be used, but no less than 8 pt. in size. Authors should use the Table option of Microsoft Word to create tables. If the reviewer could specify what part of the tables need improvement, we are happy to consider these suggestions.
Reviewer 3 Report
Comments and Suggestions for Authors
This manuscript presents a thorough review of literature on feline URI and associated risk factors. The authors have clearly done a lot of work combing through the body of research and I appreciate the clear and detailed description of the methods used. This report summarizes key findings across a variety of studies in a concise way that will be meaningful to the practitioner. Well done!
Three over-arching points that I would ask the authors to consider in further revisions are (1) be sure not to add to the existing confusion regarding measures of frequency by inconsistent or redefined use of the term “prevalence,” (2) to clarify and perhaps re-frame some of the take-home recommendations and their utility (ie, are these recommendations to improve additional epidemiological research or to improve the applicability of clinical research questions to patient care), and (3) review citations to be sure that the concepts attributed to the citation are representative of what was actually studied/reported.
See the following points for more specific comments and suggestions:
Line 57 – Typo: ...due to...
Lines 58-60. This sentence feels a little out of place without any data on the rates of URI. If this sentence feels important to include, consider adding some commentary to tie this point to the previous discussion – even if it is just that in the author’s experience in this specific shelter/region/ shelter type, most of the medical or infectious euthanasia of cats is due to URI.
Line 105-106. Shouldn’t the research question match that described in lines 109-113? That is, any measures of frequency (as defined later) of URI were assessed, not just the prevalence.
Line 159. Similar to previous comment – is prevalence the correct word here?
Lines 175-177. I appreciate the clear definitions provided here, though I question describing or interpreting terms that included the word “rate” as prevalence. Would not the majority of these be reports of incidence rather than prevalence? As you will go on to present, time (or “length of stay” for the shelter studies) is an important risk factor so this seems like an important distinction for this particular disease, perhaps moreso than others. It may be more clear and accurate to avoid using the word prevalence in this way in the actual manuscript even if these measures are functionally grouped together (ie, only say “prevalence” when you actually mean prevalence or it is the term that was used in a reviewed study – as described in Table 4). At minimum, consider further explanation of these terms, the distinction and importance for URI, and the decision to group these measures together (probably best for the Discussion).
Lines 200-201/Figure 3. Does this imply that the populations of animals studied lived in these regions? The wording here, particularly on the caption to Figure 3, is a little vague. Location of the populations would be a more useful parameter than the location of the author or their affiliation.
Lines 341-344. Consider rewording/softening this phrase. The general point is valid however, as written, the implication is that these studies chose a poor measure of estimating disease prevalence, while estimating disease prevalence was not actually the objective of these reports.
Line 373. Is a subheading missing for this paragraph? Or is this meant to be in one of the subsections below?
Lines 415-417. Yes, and many (most?) cats come into shelters already having acquired the pathogens that lead to these diseases. It is the stress of living in a shelter and the increasing stress with increased LOS that often (and possibly more importantly) lead to upper respiratory disease. Consider rephrasing and putting additional context to the multifactorial nature of the shelter-related risk factors.
Lines 422-424. I’m not sure this is exactly correct. Quarantine facilities are for animals with known (or in some cases high risk of) previous exposure to specific pathogens. An additional and important purpose of quarantine is to identify those animals that do go on to develop fulminant disease so that they can be managed appropriately.
Line 429-430. Cleaning or disinfection or both?
Lines 457-459. I don’t disagree with this recommendation though find it a little confusing as grouping different studies together and eliminating any distinctions was exactly the process followed to create this report.
Lines 459-461. Again, this would depend on the objective of the specific study. And while there is a lot of overlap, there are a number of distinct clinical signs that can allude to a particular pathogen and have meaningful clinical and operational consequences (eg, lingual ulceration often implies calicivirus, corneal ulceration often suggests herpesvirus).
Lines 462-463. As you have described throughout this report, this is indeed done in many (most/) cases.
Line 465. See previous comments on use of prevalence and variation in study objectives.
Line 468-470. I understand the recommendation to focus on environmental factors as these have been shown to have an impact, but am confused by the apparent recommendation to limit these to those listed here. How will additional environmental factors be discovered if the focus remains only on the known?
Lines 475-479. I don’t believe the yield of PCR tests vs other methods or comparison of recovery rates via different swab sites were evaluated in the current report, merely the proportion of studies that used each methodology was described. Please clarify and/or reference.
Author Response
We thank the reviewer for their detailed comments and appreciate the time and effort taken over this peer review. Individual comments are addressed below:
- Be sure not to add to the existing confusion regarding measures of frequency by inconsistent or redefined use of the term “prevalence,”: Diversity of terminology was one of the more significant findings in our review. To ensure that we do not commit the same mistakes of using various terms to mean different things, we have pre-defined our terminology in lines 175-180. We hope that this will eliminate any confusion while reading our review, both, during interpretation of studies and when we have quoted studies by using the same terminology that they used. "For consistency within our review, where studies have conducted laboratory testing and used terms such as ‘test positivity’, ‘pathogen shedding rate’, ‘detection rate’ and ‘recovery rate’, we interpret them to mean ‘prevalence’. Amongst studies that did not conduct laboratory tests, we define the term ‘prevalence’ for the number or percent of animals with clinical signs. ‘Clinical signs’ denotes presence of visible ocular, nasal or other respiratory signs that indicate likelihood of URI infection."
- To clarify and perhaps re-frame some of the take-home recommendations and their utility (ie, are these recommendations to improve additional epidemiological research or to improve the applicability of clinical research questions to patient care): The recommendations we make are mainly for researchers attempting to conduct Systematic Reviews on FURTI in the future (lines 452-470). Where risk factor analyses cover a large range of environmental predictors, we recommend focussing on synthesis of data for risk factors that are modifiable in a practical sense, rather than factors that are out of control of animal carers (eg. housing density vs. seasonal variation). This has been added to the relevant section.
- Review citations to be sure that the concepts attributed to the citation are representative of what was actually studied/reported: We thank the reviewer for this comment, and we have tried to address this where specifically identified by the reviewer.
- Line 57 – Typo: ...due to...: This has been corrected.
- Lines 58-60. This sentence feels a little out of place without any data on the rates of URI. If this sentence feels important to include, consider adding some commentary to tie this point to the previous discussion – even if it is just that in the author’s experience in this specific shelter/region/ shelter type, most of the medical or infectious euthanasia of cats is due to URI: We have added the following sentence to this section: "While we were unable to extract data on specifics of these medical/infectious reasons, it has been the authors’ experience that many of these were due to Feline URT infections, especially where the clinical picture was further complicated by concurrent comorbidities (skin infections, chronic systemic diseases, FIV status)."
- Line 105-106. Shouldn’t the research question match that described in lines 109-113? That is, any measures of frequency (as defined later) of URI were assessed, not just the prevalence: We have modified these sentences to the following: "While our review does not include bias assessment or complete synthesis of research findings, it provides a comprehensive description of the literature on frequency data reported for feline URI in multiple contexts and explores whether the literature lends itself to be systematically synthesised. Our objective was to conduct a scoping review of observational studies on feline URI. The research question was: “What is the current body of evidence on the presence of feline URI and its association with potential risk factors? Can this information be systemically synthesised?”"
- Line 159. Similar to previous comment – is prevalence the correct word here?: We have changed the term to "frequency data".
- Lines 175-177. I appreciate the clear definitions provided here, though I question describing or interpreting terms that included the word “rate” as prevalence. Would not the majority of these be reports of incidence rather than prevalence? As you will go on to present, time (or “length of stay” for the shelter studies) is an important risk factor so this seems like an important distinction for this particular disease, perhaps moreso than others. It may be more clear and accurate to avoid using the word prevalence in this way in the actual manuscript even if these measures are functionally grouped together (ie, only say “prevalence” when you actually mean prevalence or it is the term that was used in a reviewed study – as described in Table 4). At minimum, consider further explanation of these terms, the distinction and importance for URI, and the decision to group these measures together (probably best for the Discussion): We appreciate the reviewer's perspective, and it is one we share. However, we found that many studies used the word "rate" inappropriately. For example, Aziz et al. (2018) uses "URI rate" and "prevalence rate" to report point estimates as %. They also state that they are reporting on "infectious disease prevalence" and would have liked to also report on "incidence rates" for a "more comprehensive understanding", so they do understand the difference between a point estimate (like prevalence) and a temporal measure (like incidence rate). For our review, to enable reporting on such studies, we have interpreted their point estimates as true prevalence measures yet acknowledged that the terminology they used includes the word "rate". We hope this is sufficient explanation for our readers.
- Lines 200-201/Figure 3. Does this imply that the populations of animals studied lived in these regions? The wording here, particularly on the caption to Figure 3, is a little vague. Location of the populations would be a more useful parameter than the location of the author or their affiliation: The analysis only looks at where the cat populations were resident, and not at author affiliations. We have added "animal populations" to this sentence to clarify.
- Lines 341-344. Consider rewording/softening this phrase. The general point is valid however, as written, the implication is that these studies chose a poor measure of estimating disease prevalence, while estimating disease prevalence was not actually the objective of these reports: We agree with the owner and have softened the sentence. We do find that at least one of the studies cited had as its aim "to determine the seroprevalence of FHV and FCV" and was using antibody titres to "determine extent of circulating pathogens" (Dall'ara 2019), however we appreciate that the motivation for the study was not simply to report on prevalence but to use seroprevalence as a means of effective vaccination of susceptible populations.
- Line 373. Is a subheading missing for this paragraph? Or is this meant to be in one of the subsections below?: We apologise- this paragraph belonged to a deleted section and has now been removed.
- Lines 415-417. Yes, and many (most?) cats come into shelters already having acquired the pathogens that lead to these diseases. It is the stress of living in a shelter and the increasing stress with increased LOS that often (and possibly more importantly) lead to upper respiratory disease. Consider rephrasing and putting additional context to the multifactorial nature of the shelter-related risk factors: The sentence has been modified to: "The longer an animal stays in a shelter, the greater its stress and exposure to pathogens, and subsequent chances of getting infected with common endemic diseases".
- Lines 422-424. I’m not sure this is exactly correct. Quarantine facilities are for animals with known (or in some cases high risk of) previous exposure to specific pathogens. An additional and important purpose of quarantine is to identify those animals that do go on to develop fulminant disease so that they can be managed appropriately: We believe we have stated the same as the reviewer: "Quarantine.....based on the assumption that incoming animals are likely to be carrying infectious diseases". This would be through previous exposure to specific pathogens of particular interest.
- Line 429-430. Cleaning or disinfection or both?: Modified to "cleaning and disinfection.
- Lines 457-459. I don’t disagree with this recommendation though find it a little confusing as grouping different studies together and eliminating any distinctions was exactly the process followed to create this report: This report had a broad purpose of examining all of the observational and epidemiological literature on this disease, to determine if data could be pooled together for a meta-analysis. Since we found that diversity of studies precluded this, we recommend that for a potential systematic study, to enable pooling of data, the focus should be narrowed to either only studies reporting on clinical disease or only studies reporting on presence of specific pathogens.
- Lines 459-461. Again, this would depend on the objective of the specific study. And while there is a lot of overlap, there are a number of distinct clinical signs that can allude to a particular pathogen and have meaningful clinical and operational consequences (eg, lingual ulceration often implies calicivirus, corneal ulceration often suggests herpesvirus): We agree with the reviewer. We have clarified this sentence to reflect our meaning better: "For studies synthesizing the prevalence of clinical feline URT disease, a clear definition of the disease syndrome should be stated, and include all of the commonly seen signs such as nasal discharge, ocular discharge, conjunctivitis, corneal ulceration, coughing, sneezing and oral ulceration."
- Lines 462-463. As you have described throughout this report, this is indeed done in many (most/) cases: That is correct. However, this sentence refers to when a data synthesis is carried out (rather than reporting on individual populations) in the future. We have clarified this in this sentence.
- Line 465. See previous comments on use of prevalence and variation in study objectives: This has been corrected. "Prevalence measures" has been changed to "frequency data".
- Line 468-470. I understand the recommendation to focus on environmental factors as these have been shown to have an impact, but am confused by the apparent recommendation to limit these to those listed here. How will additional environmental factors be discovered if the focus remains only on the known?: This has been modified, to say "These include (but are not restricted to)".
- Lines 475-479. I don’t believe the yield of PCR tests vs other methods or comparison of recovery rates via different swab sites were evaluated in the current report, merely the proportion of studies that used each methodology was described. Please clarify and/or reference.: The reviewer is correct- the agreement on PCR vs other methods was within individual studies and not formally evaluated in our review. This sentence has been modified to: "While several individual studies found that PCR tests yield higher positivity rates when compared to other diagnostic methods such as viral or bacterial isolation, the relevance of a positive PCR test to clinical signs warrants further research."
Reviewer 4 Report
Comments and Suggestions for Authors
A scoping review of the literature on feline upper respiratory tract infections was carried out in lieu of a systematic review because of the heterogeneity in study approaches for this condition. Overall the paper is well written and structured, and the methodology appears sound.
There is a relatively large amount of diverse data to analyse and draw together from very different study designs with varying limitations. They have made a good attempt to do this, and have identified some risk factors common to several studies such as age, environmental factors in shelters and length of stay. A number of these risk factors have been fairly well established for many years, but nevertheless it is useful to see some of the available data collated in this type of review.
The discussion attempts to draw the key findings together and suggests the way forwards for future studies, and this is done reasonably well. The authors clearly have a good understanding of most aspects of the syndrome, though the basics of the epidemiology, transmission and control of the disease with the different pathogens could perhaps be explained a bit more. There are also some instances when the reference given is not quite appropriate – though perhaps not surprising in a large review of this nature.
Author Response
We thank the reviewer for their comments. We have addressed the one comment that suggests modifications below:
The authors clearly have a good understanding of most aspects of the syndrome, though the basics of the epidemiology, transmission and control of the disease with the different pathogens could perhaps be explained a bit more: We agree with the reviewer, however due to the limitations of word count and the scope of this review, we feel that these topics were only able to be covered briefly. There are many peer reviewed articles as well as textbooks that cover the more biological aspects of this disease in great depth, and we have cited some of them in our review for any reader who wishes to learn more about transmission, pathogenesis and control of this disease in various settings.
Reviewer 5 Report
Comments and Suggestions for Authors
Feline upper respiratory infection is a severe problem in cat populations and this work is a fine review on the epidemiology and related risk factors to define guidelines.
Author Response
We thank the reviewer for their comments.